# Novel Treatment Strategy for Patients with Venom-Induced Consumptive Coagulopathy from a Pit Viper Bite

**DOI:** 10.3390/toxins12050295

**Published:** 2020-05-05

**Authors:** Eun Jung Park, Sangchun Choi, Hyuk-Hoon Kim, Yoon Seok Jung

**Affiliations:** Emergency Department, Ajou University School of Medicine, Suwon 16499, Gyeonggi-do, Korea; amita62@nate.com (E.J.P.); hyukhoon82@gmail.com (H.-H.K.)

**Keywords:** snake venoms, consumption coagulopathy, thromboelastography, antivenins, blood transfusion

## Abstract

Pit viper venom commonly causes venom-induced consumptive coagulopathy (VICC), which can be complicated by life-threatening hemorrhage. VICC has a complex pathophysiology affecting multiple steps of the coagulation pathway. Early detection of VICC is challenging because conventional blood tests such as prothrombin time (PT) and activated partial thromboplastin time (aPTT) are unreliable for early-stage monitoring of VICC progress. As the effects on the coagulation cascade may differ, even in the same species, the traditional coagulation pathways cannot fully explain the mechanisms involved in VICC or may be too slow to have any clinical utility. Antivenom should be promptly administered to neutralize the lethal toxins, although its efficacy remains controversial. Transfusion, including fresh frozen plasma, cryoprecipitate, and specific clotting factors, has also been performed in patients with bleeding. The effectiveness of viscoelastic monitoring in the treatment of VICC remains poorly understood. The development of VICC can be clarified using thromboelastography (TEG), which shows the procoagulant and anticoagulant effects of snake venom. Therefore, we believe that TEG may be able to be used to guide hemostatic resuscitation in victims of VICC. Here, we aim to discuss the advantages of TEG by comparing it with traditional coagulation tests and propose potential treatment options for VICC.

## 1. Introduction

Pit vipers inhabit Asia, North America, Europe, and Africa [1]. Envenomation through a pit viper bite is frequently reported, and it can lead to critical illnesses [1,2]. Venom-induced consumption coagulopathy (VICC) is a central issue following a pit viper bite because it can be lethal [2]. However, it is currently difficult to characterize coagulopathy and identify specific treatment because of the complex pathophysiology of coagulopathy following a pit viper bite [1].

Recently, thromboelastographic analysis of hemostasis has been performed to elucidate the pathophysiology of coagulopathy in patients after a pit viper bite [3,4,5,6,7,8]. In this review, we aim to discuss the advantages of thromboelastography (TEG) by comparing it with traditional coagulation tests to identify the potential treatments for VICC.

## 2. Characteristics of Envenomation by a Pit Viper

Unlike poisoning associated with other medications and toxic materials, envenomation from a snake has specific clinical attributes. First, snake venom is composed of various toxins [4,9]. Venoms are generally composed of several enzymatic and nonenzymatic proteins and peptides [1]. Snake venoms have different compositions depending on the family and species the snake belongs to [4,9]. Different toxins lead to different clinical manifestations such as neurotoxicity, myotoxicity, or coagulotoxicity [3]. Venoms of different snake species have different clinical toxicity levels [9]. Pit vipers in different regions, despite belonging to the same species, may show distinct envenomation characteristics [3,5,9]. *Gloydius intermedius* and *G. ussuriensis* in Korea and Russia have shown markedly different effects on coagulotoxicity [4]. However, the antivenom from one or two species in one local area can effectively neutralize envenomation from a species from a different local area [10]. Second, the dose of delivered venom is a factor that can be used to determine the degree of envenomation [1]. The amount of injected venom can be controlled based on the purpose of a snake bite [1]. A dry bite, that is, the bite without venom infusion, does not cause envenomation, whereas an offensive bite can cause severe toxicity because of the delivery of a large dose of venom [1]. Third, the species of the victim is a determining factor of the characteristics of envenomation [1]. Bite sites and the victim’s systemic response to the toxin can lead to different type of envenomation manifestation [1,9]. Fourth, the bite of a single viper species may induce different clinical manifestations in different species [1,3,9]. A bite by *Calloselasma rhodostoma* (Malayan pit viper) may cause thrombosis in a small animal, while it may induce hemorrhages in larger animals such as humans [3].

Clinical manifestations are commonly categorized into systemic symptoms and localized symptoms [1]. Systemic symptoms are diverse, ranging from nausea and dizziness to critical manifestations, including hypotension, neurologic deterioration, coagulopathy and organ failure [1]. The presence and severity of systemic symptoms usually lead to the use of antivenom [1,2,10,11].

## 3. VICC

Of the systemic symptoms associated with envenomation, coagulopathy induced by pit viper venom has unique characteristics [2]. The coagulopathy is caused by the inhibition of coagulation factors and the consumption of coagulation factors secondary to the promotion of coagulation [2]. This characteristic is called venom-induced consumptive coagulopathy (VICC), with a pathway involvement similar to that of disseminated intravascular coagulopathy (DIC) [2]. VICC frequently occurs following Asian viper envenomation [4]. Venomous snakes in Korea cause VICC, which has a prevalence of 14–50% [12,13,14].

### 3.1. Pathophysiology of VICC

VICC affects multiple steps of the coagulation pathway, i.e., the toxins exert an influence on the platelets, coagulation factors, coagulation products, and blood vessels [2]. Toxins can be classified into two different groups on the basis of the effects of procoagulant and anticoagulant use [2,3,5,15].

Procoagulant in the venom activates coagulation factors, leading to the formation of factor Ia (fibrin) [2,3,4]. The venom activates factor Ⅹ and factor II (prothrombin), resulting in the creation of factor IIa (thrombin), and the fibrin generated in this process has stable bonds [2,3,4]. However, the effects of procoagulants vary greatly according to the species of snakes [3]. *C. rhodostoma* and *Hypnale hypnale* (hump-nosed pot viper) have procoagulants. However, *C. rhodostoma* activates factor X, whereas *H. hypnale* does not [3]. Although procoagulants in venoms may induce thrombosis or embolic events, they usually lead to the consumption of coagulation factors, which may cause severe spontaneous bleeding [2].

Anticoagulants present in the venom can be classified into two groups depending on the manner in which they affect the coagulation pathway [4]. One type of anticoagulant directly inhibits the production of coagulation factors, especially factors Ⅹa, Ⅸa, and ⅩIa, and thrombin [3,4]. The other type of anticoagulant causes fibrinogenolysis [2,3,4]. Fibrinogenolysis is caused by the direct destruction or the consumption of fibrinogen [3,4,5]. Fibrin activated by anticoagulants present in venom is weak and short-lived [3,4,5]. This kind of fibrin will not enable the production of factor ⅩIII, resulting in the failure of fibrin polymerization [5]. As the fibrin without polymerization is unstable and can be degraded easily, the breakdown of fibrinogen and fibrinolysis leads to a net-anticoagulation effect. This type of anticoagulant, also called a pseudo-procoagulant, can be categorized in the procoagulant group [2,3,4,5].

Procoagulants and anticoagulants present in venom eventually induce VICC [2]. Considering that both components are usually found in analyzed venom, the characteristics of VICC due to bites form certain vipers is affected by venom composition and the timing of toxin onset [2,3,4,5].

### 3.2. Clinical Manifestation of VICC

VICC can cause bleeding with various levels of severity [1,10,14,16]. Uncontrolled bleeding from the bite site or hemorrhagic bullae present as a severe form of localized symptoms [1]. In critical cases, gastrointestinal bleeding, hemorrhagic infarction of the brain, or multiple organ failure similar to the symptoms in DIC, has been noted [17,18,19,20,21,22]. However, multiple organ failure or clinical deterioration is less commonly observed in VICC than in DIC caused by other factors [2,10,19,23]. Most patients with VICC show abnormal laboratory findings [2,6,7,8,10,19,23,24,25,26].

The laboratory findings of VICC patients include thrombocytopenia, prolonged coagulation time (clotting time, prothrombin time [PT], and activated partial thromboplastin time [aPTT]), decreased fibrinogen levels, increased fibrinogen degradation product levels (FDP, XDP, and D-dimer), decreased coagulation factor levels, and decreased antithrombin III levels [2,6,7,8,10,14,19,24,25,26,27,28,29,30]. Initial laboratory results are usually within normal ranges [2,7,8,25,26]. The latency of laboratory coagulopathy has been reported to be two days to two weeks [14,28,29,31]. However, the severity of laboratory findings are not necessarily correlated with the severity of clinical manifestations; however, laboratory findings partially mirror the clinical manifestations [2]. Therefore, there is a need for a preemptive index or detection method that predicts the changes in the progress of injury during the latent period.

### 3.3. Diagnosis of VICC

#### 3.3.1. Limitations of Traditional Coagulation Tests

Traditionally, whole blood clotting time (WBCT), PT, and aPTT have been used to diagnose coagulopathy in VICC [2]. WBCT is a measure of the time of clotting in a patient’s blood for when laboratory tests are unavailable [2]. The 20-min WBCT, which is a refined form of WBCT, is a simple and rapid test to determine the presence of coagulopathy [2,27,32]. However, abnormal findings in 20-min WBCT were only observed in 1–39% of patients according to previous studies, and, it is not correlated with the patients’ clinical manifestations [7,27]. As the 20-min WBCT cannot indicate the cause of coagulopathy in VICC concisely, coagulation tests of a greater scope are recommended [2].

Tests to assess PT and aPTT are available in most facilities, and these parameters can be easily computed and analyzed [2,32,33,34]. According to Pongpit et al. [32], PT and aPTT are excellent diagnostic markers for VICC. However, failure to detect VICC in the early stages on the basis of PT and aPTT has been reported in other studies [7,25,26]. Initially, the PT and aPTT levels of VICC patients are within the normal range, but these factors eventually change and become abnormal over a few days, which suggests the occurrence of VICC. The use of PT and aPTT alone for diagnosing coagulopathy has many limitations [7,25,26,33,34]. First, consumptive coagulopathy from a hemorrhagic cause such as VICC can be influenced by coagulation, fibrinolysis, and mechanical factors [33]. PT and aPTT can be affected by these factors but do not help determine the specific cause of VICC. Second, PT and aPTT do not show in vivo interaction between the platelets and coagulation factors [33]. Third, PT and aPTT are unable to analyze the stability of thrombin that is formed by polymerization of fibrin by factor ⅩⅢ [33,35]. PT and aPTT are usually obtained in 10 min, before the polymerization of fibrin occurs. Fourth, PT and aPTT cannot reflect the formation of thrombin that lacks antithrombin or protein C [33,35]. Finally, PT and aPTT are unable to detect the presence of fibrinolysis [2,33]. With these limitations, PT and aPTT do not fully reflect VICC, which is associated with a depletion of coagulation factors, fibrinolysis due to the formation of the fragile fibrin clots, or the dysfunction of platelets.

Platelet count, fibrinogen levels, and fibrin degradation products have also been analyzed for the diagnosis of VICC [2]. However, these tests only show some specified factors and require additional tests to diagnose VICC [2].

#### 3.3.2. Thromboelastography

Thromboelastography (TEG) has been widely used during transfusion in patients with trauma as well as during surgeries. [35,36]. A TEG analyzer consists of a torsion wire, a pendulum, and a disposable cup to hold the sample of whole blood. The cup is rotated back and forth through an angle of 4.45°, six times a minute, to imitate sluggish venous flow, for the whole duration of the test. A pin attached to the torsion wire is inserted into the cup with the whole blood sample. As the coagulation cascades progress, the pin produces electrical signals. Typical waveforms regarding the speed and strength of clot formation are displayed as signals in TEG, and the measurements are transferred onto the computer [35,36].

TEG is advantageous over traditional coagulation analysis. First, TEG is less time consuming than analysis of coagulation factors [34,36]. The laboratory tests for platelets, PT, and aPTT usually take a shorter time and can be carried out in most facilities. However, fibrinogen and fibrin degradation products take a lot longer to analyze and testing for these factors may be impossible in some facilities. TEG takes 30–60 min in general and can take 30–40 min when testing samples without clotting [36]. Some facilities perform TEG as a point-of-care test (POCT) to decide whether resuscitation should be performed during emergency situations [25,35,36]. However, frequent and precise quality control is necessary for TEG POCT [35,36]. Second, information that cannot be obtained from PT and aPTT can be gathered [34,35]. Fibrin strength can be measured and represented as the maximal clot firmness (MCF) or maximal amplitude (MA) [33,35,36]. MCF and MA values smaller than the control values indicate a reduction in fibrin polymerization or weak polymerization [33,35,36]. On the basis of this finding, the pseudo-procoagulant effect can be detected in VICC. MCF and MA are possibly reduced by thrombocytopenia or platelet dysfunction [35,36]. After the activated platelet with the GpIIb/IIIa receptor interacts with fibrinogen, platelet aggregation occurs [35,36]. Platelets without proper receptors, cannot form a strong bond, resulting in a reduction in MCF and MA. As TEG ends after the polymerization of fibrin, the production of thrombin can be estimated [33]. However, platelet, fibrinogen, and factor ⅩⅢ levels highly affect thrombin formation [33,35,36]. Thrombin production can still be detected, assuming that the levels of these factors are within the normal range. Fibrinolysis can also be detected [32,33,35,36]. Typically, clot lysis after 30 min (LY 30) or lysis time (LT) can be evaluated 30 or 60 min after MA [36]. A >15% decrease in CL and LT from MCF and MA indicates fibrinolysis [35]. Moreover, a hypercoagulable state may be estimated on the basis of the normal platelet and fibrinogen levels [35,36].

The utility of TEG in diagnosing and treating VICC, has been discussed in previous studies from various countries, including the USA, South Africa, India, and Japan [6,7,8,24,25,26]. Early stages of VICC can be determined using TEG, even if the PT and aPTT remain normal [7,25,26].

After a few days, all of these cases showed prolongation of PT and aPTT [7,25,26]. Hadley et al. reported that TEG had a higher sensitivity than that of the international nationalized ratio (INR) in detecting severe VICC [6]. Moreover, TEG may be able to detect an improvement in VICC faster than PT [24].

TEG may be able to detect VICC presenting with procoagulation and fibrinolysis by degradation or fragile fibrin formation. In previous studies that analyzed coagulopathy using TEG, VICC was found to have mixed procoagulant and anticoagulant characteristics [3,4,5]. Although the effects of procoagulants or anticoagulants varied in every case, VICC is generally associated with the occurrence of net-anticoagulation effects [3,4,5]. *C. rhodostoma* and *H. hypnale* have strong procoagulant components and can decrease the clotting time [3]. However, they also have pseudo-procoagulant effects, which can weaken fibrin and thus lead to an anticoagulation state [3]. With the procoagulant effect, early initiation of coagulation and preservation of clot strength can be observed (Figure 1A) [5]. Because of the anticoagulant effect, which inhibits the factors or the direct destruction of fibrinogen, delayed initiation of coagulation and decreased strength of the clot may be detected (Figure 1B). With the emergence of species that can cause severe fibrinogen destruction such as *G. intermedius, G. brevicaudus,* and *G. ussuriensis* in Korea, no coagulation is detected on basic TEG [4]. With a pseudo-procoagulant, early initiation of coagulation and a decrease in the strength of the clot may be shown (Figure 1C) [4,5]. With fibrinolysis, the strength of the clot may decrease by >15% after 30 min (Figure 1D) [5].

Controlled transfusion has been studied in several situations using TEG [33,35,36,37]. These studies included patients who experienced trauma and surgeries, but none of them reported on VICC [33,35,37]. Severe hemorrhage can lead to DIC with the consumption of coagulation factors, which usually leads to massive transfusion [35,36,37]. To reduce consumptive transfusion, some studies have used a transfusion algorithm or principles concerning transfusion or medication as indicated by TEG [30,31,32]. In several situations such as cardiac surgery or liver transplantation, using TEG has been effective in reducing transfusion but has no impact on patient’s outcome [37]. However, TEG may have the ability to detect the development of VICC. Hence, we suggest that TEG should be incorporated during the hemostatic resuscitation of patients with VICC.

There are several obstacles to the clinical use of TEG. First, TEG is a relatively expensive infrastructure and a high-cost test [24]. Therefore, it is impossible to perform this method in resource-poor settings, where most venomous toxic snake bites occur. Second, no randomized controlled study has examined the effectiveness of TEG in patients with VICC. Thus, its efficacy remains undetermined. Further studies to evaluate the efficacy of TEG in VICC are needed. Third, quality controls for the POCT apparatus are frequently needed to achieve rapid, accurate, and precise results [35,36].

#### 3.3.3. Immunological Test

Identifying the species of snake can enable the use of monovalent antivenom in some regions [38,39]. However, identifying the species of the snake on the basis of the information from a witness is difficult [39]. Thus, polyvalent antivenom is commonly administered in patients with snakebites who need an antivenom. Immunological tests such as enzyme immunoassay or enzyme-linked immunosorbent assay are used to determine the characteristics of the venom [39]. Most immunological tests are time consuming and hence have limited utility in urgent clinical settings [39]. However, the Snake Venom Detection Kit (SVDK) has been used for the quick identification of venom and has shown efficacy in determining monospecific antivenom in Australia, India, and Vietnam [38,39,40].

However, the use of immunological tests and kits has limitations. The first limitation is associated with its high cost [39]. Second, high false-positive rates and cross-reactivity can lead to incorrect results [39,41]. Third, to increase the detection rate, sampling by bite-site swab is recommended for SVDK [38,39,40]. However, detection of venom at the bite-site is not correlated with systemic envenomation [39,40]. If a bite-site swab is unavailable, a urine sample can be used, but urine produced before a bite can render a false-negative result [38,40]. Fourth, after the detection of the specific venom, a monospecific antivenom may still be unavailable. For example, there are three kinds of venomous *Gloydius* species in Korea [10]. However, only one type of monospecific antivenom is available for Chinese *Agkistrodon Halys* in South Korea [10].

### 3.4. Treatment of VICC

#### 3.4.1. Antivenom

The traditional treatment of VICC is the administration of antivenom. If VICC is suspected, antivenom is strongly recommended [1,2,10,13,23,38]. Antivenom comprises immunoglobulin that binds to and neutralizes toxins [10]. If the laboratory abnormalities continue or clinical manifestations of VICC worsen for > 6 h after antivenom therapy, additional antivenom administration is recommended [1,2,10,23,38]. However, the use of antivenom does not merely depend on the deterioration of the patient’s clinical condition or abnormalities in the patient’s laboratory tests [42]. Dose, time, and frequency of additional antivenom administration can be guided by monitoring the changes in the coagulation status using TEG [26].

When using antivenom, some aspects should be taken into consideration. Antivenoms can neutralize procoagulants promptly, but the restoration of the consumed coagulation factors may take 1 to 2 days [38,43]. The deficiency of coagulation factors may persist after the administration of antivenom. After a sufficient dose of antivenom is administered, supplementation of coagulation factors has been proposed in some studies [38,43]. As mentioned above, toxins can vary even among the same species from different regions [4,5,9]. A SVDK can be used to determine the specific type of venom [38,39,41]. However, mono-specific antivenom may be unavailable in some regions, hence, it may be inefficient to make antivenoms that are specific to a certain species in a region [2,10]. Poly-specific antivenom can be used because pit-vipers have common antigens and cross-reactivities, according to previous studies [10,11,44,45]. Another limitation is an allergic reaction to antivenom [1,2,23,46]. The symptoms may be mild, such as skin rash, pruritus, and vomiting, or severe, such as hypotension, bronchospasm, or angioedema [2,23,46]. Serum sickness can occur several days after administration [23,47]. The scarcity of antivenom in some regions is also one of the limitations of antivenom therapy [2,10,23].

#### 3.4.2. Transfusion

As patients with VICC show abnormalities in coagulation tests such as DIC, those who underwent blood transfusions frequently presented with abnormal laboratory results [2,13,14]. Previous literature has indicated that transfusion in patients with VICC can be dangerous [2,10,23] because blood transfusion may further trigger consumptive coagulopathy [2]. Additionally, transfusion without antivenom therapy should cause more complications than transfusion with antivenom therapy [13]. Patients with VICC showed severe abnormalities in laboratory tests, which do not correspond to the clinical manifestations reported in previous studies [2]. Without severe hemorrhage, transfusion for correction of laboratory findings is not recommended [2,23,48]. Unlike these studies, other studies have reported that transfusion of coagulation factors may be effective in patients with VICC [38,43]. In these studies, administration of antivenom followed by fresh frozen plasma (FFP) within 4 h shows a rapid resolution of coagulation abnormalities, but it has no effect on patient’s outcomes [38,43]. Venomous snakes in Korea also tend to produce VICC; FFP or cryoprecipitate with antivenom is administered to treat severe coagulopathy [13]. Therefore, limited transfusion may be the key to supplying coagulation factors and controlling VICC [49]. The type and dose of transfusion can be guided by monitoring the changes in coagulation status on TEG. On the basis of the results of the previous studies regarding the use of the TEG-based transfusion algorithm [33,49,50,51], FFP transfusion can be considered when the R time is >10 min. Furthermore, platelet transfusion can be considered when the MA is <50 mm. LY 30 >3% is critical for the initiation of antifibrinolytic therapy, where one can consider administering an antifibrinolytic agent such as ε-aminocaproic acid or tranexamic acid in patients with VICC [33,49,50,51]. With post-antivenom TEG, controlled transfusion may be conducted effectively in patients with VICC.

Further explanations for the proposed treatment algorithm in VICC due to a pit viper bite (Figure 2) will now be described. Adding TEG to traditional coagulation tests can lead to early detection of VICC. Identification of venom and subsequent administration of monospecific antivenom is the most effective and desirable treatment. Performance of a rapid immunoassay such as the SVDK along with traditional diagnostic workup may help identify the snake species correctly. However, such an immunoassay kit was used only in limited regions of several developed countries and in circumstances in which the said kit was available. Thus, the use of an immunoassay kit could be omitted depending on its availability. Poly-specific antivenom would be administered if the venom identity is unclear or the spectrum of species is broad. Dose, time, and frequency of antivenom administration can be guided by monitoring TEGs. In addition, adding TEG to traditional coagulation tests can influence the decision to use a transfusion and can aid in controlling the dose of transfusion. Clinical and laboratory examinations should be repeated after 4–6 h and treatment with antivenom and transfusion should be repeated depending on the clinical progress of the individual with VICC.

## 4. Conclusions

The efficacy of snake envenomation in viscoelastic monitoring remains poorly understood. As discussed in this review, TEG can help classify procoagulant and anticoagulant effects, and evidence concerning its efficacy in managing snake envenomation is growing. Thus, TEG may have the potential to aid in the identification of the development of VICC from a pit viper bite and in monitoring the progression of a coagulopathy according to the treatment. Further randomized, controlled studies are needed, in particular to determine the efficacy of TEG in guiding the treatment of VICC.

## Figures and Tables

**Figure 1 toxins-12-00295-f001:**
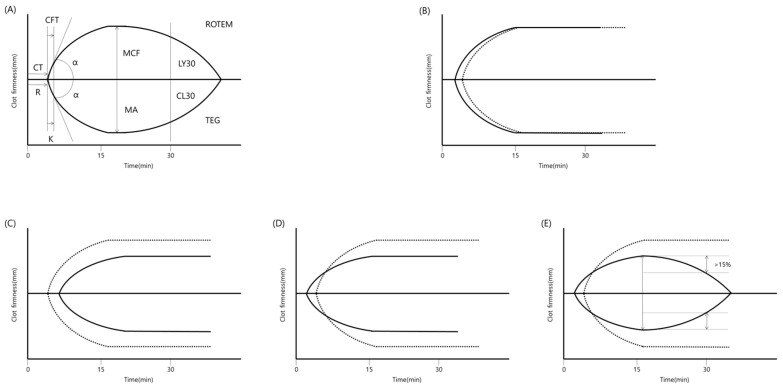
Schematic examples of thromboelastography (TEG). (**A**) Typical waveform of TEG. Terms in the upper wave are used in ROTEM and terms in lower wave are used in TEG. (**B**) Procoagulant effect with decreased delay of coagulation initiation and preservation of clot strength. (**C**) Anticoagulant effect with increased delay of coagulation initiation and decreased clot strength, (**D**) Pseudoprocoagulant effect with decreased delay of coagulation initiation and decreased clot strength, (**E**) Fibrinolysis effect with a delayed decrease in the clot strength by >15%. The dotted line indicates the normal presentation. CT, coagulation time, CTF, clot formation time, MCF, maximal clot firmness, α, slope between CT and CFT, LY30 or R and K, lysis after 30 min, R, time of latency to initial fibrin formation, K, time to get amplitude of 20 mm, MA, maximal amplitude, CL30, clot lysis after 30 min.

**Figure 2 toxins-12-00295-f002:**
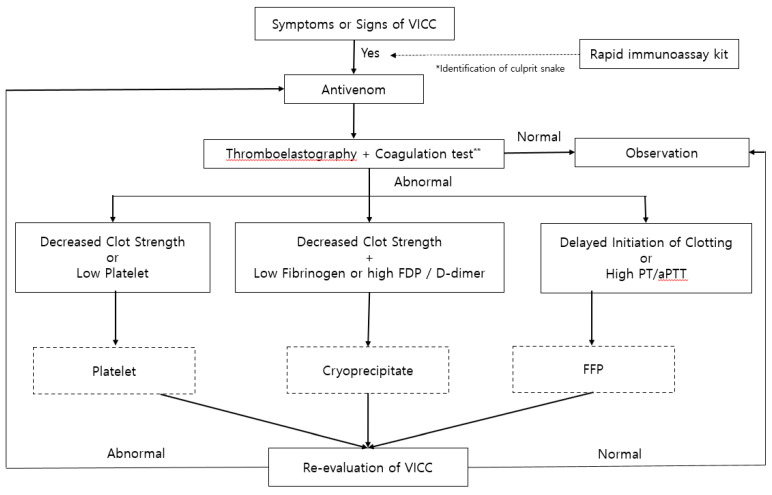
Proposed treatment algorithm in venom-induced consumptive coagulopathy following a pit viper bite. Dotted boxes indicate the optional treatments according to the results of laboratory tests and the clinician’s opinion. Grey boxes indicate the additional tests and potential treatments based on the results. **Coagulation test includes prothrombin time, activated partial thromboplastin time, fibrinogen, fibrinogen degradation product, and D-dimer.

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
