# Peer review of "Novel Treatment Strategy for Patients with Venom-Induced Consumptive Coagulopathy from a Pit Viper Bite"

_toxins, 2020, doi:10.3390/toxins12050295_

Round 1
Reviewer 1 Report
The authors took into account my comments and suggestions.
Author Response
N-C
Reviewer 2 Report
Line 18 - this statement is a too strong. Rather than sayin TEG can I would recommend softening the statement to "may be able to"
Line 40 - this sentence doesn't make sense
Lines 137-138 - consider mentioning the commercially available field TEG here
Line 143-144 - this line is poorly written and difficult to understand
Line 255-256 - This sentence does not make sense and is not well supported in this paragraph. More emphasis and review of the literature using FFP, platelets, etc. should be added to this paper as that is what makes it unique from the literature. The benefit of TEG is out there as well as its difference from traditional coagulation testing so the novelty of this paper is using the TEG to guide these other snake bite therapies. This is worthy of discussion but there is just not enough in this paper to make the point. Need more discussion on the use of these therapies in envenomation and establish a need for a way to guide their use and then talk about how TEG will help this more. Possibly delve more into how it is being used in transfusion medicine. There was just a very small comment towards this. Need to talk about reasons you should use a guide to using these therapies rather than just using them.
Author Response
Thank you for providing helpful comments. Your interest in this article and your constructive criticisms are much appreciated. I worked with all my coauthors in trying to address to every comment of yours, which led to a significant change from the original manuscript.
I answered all your points as below. The changes that are made in the article are highlighted as a yellow color. Please review this revised version and inform us of your decision. I am looking forward to your reply. My answer to your comment is followed by an arrow. Thank you very much for your consideration.
Line 18 - this statement is a too strong. Rather than saying TEG can I would recommend softening the statement to "may be able to"
--> We agree with you on this point. We changed this sentence as you recommended.
Line 40 - this sentence doesn't make sense
--> We removed this sentence.
Lines 137-138 - consider mentioning the commercially available field TEG here
--> We removed the related sentences.
Line 143-144 - this line is poorly written and difficult to understand
--> We agree with you on this point. We changed this sentence as below.
Unlike those, fibrinogen and fibrin degradation products take more time and may be unavailable in some facilities to test.
Line 255-256 - This sentence does not make sense and is not well supported in this paragraph. More emphasis and review of the literature using FFP, platelets, etc. should be added to this paper as that is what makes it unique from the literature. The benefit of TEG is out there as well as its difference from traditional coagulation testing so the novelty of this paper is using the TEG to guide these other snake bite therapies. This is worthy of discussion but there is just not enough in this paper to make the point. Need more discussion on the use of these therapies in envenomation and establish a need for a way to guide their use and then talk about how TEG will help this more. Possibly delve more into how it is being used in transfusion medicine. There was just a very small comment towards this. Need to talk about reasons you should use a guide to using these therapies rather than just using them.
--> We agree with your concerns. We added some sentences in 3.4.2 transfusion.
So limited transfusion may be the key to supplying the coagulation factor and controlling VICC. Type and dose of transfusion can be guided using TEG monitoring for a change in coagulation status. Based on the results of the previous studies regarding a TEG-based transfusion algorithm, one can consider transfusing FFP when R time is > 10 minutes. Further, one can consider transfusing platelets when MA is < 50 mm. LY 30 > 3% is critical for the initiation of antifibrinolytic therapy, where one can consider administering an antifibrinolytic agent such as ε-aminocaproic acid or tranexamic acid in patients with VICC. TEGs following antivenom administration have shown that controlled transfusion may be conducted effectively in patients with VICC.
Reviewer 3 Report
Lines 40 & 41: The sentence beginning with “First, snake venom…” is structurally awkward.
Lines 42 – 44: The word various has been used repeatedly; rewrite to make the concept of venom complexity clearer.
Line 47: Replace showed quiet with “have shown quite.”
Line 67 – 68: The sentence beginning with “VICC was commonly…” is structurally awkward.
Section 3.3.1: Although PT and aPTT are limited in the diagnostics you report, are they adequate to demonstrate a coagulopathy from the snake bite indicating a need for treatment? If so, wouldn’t this lead the attending physician to order further diagnostics?
Line 143: What “some test” are you referring to that requires significantly longer time for completion?
Line 147 – 148: Do facilities that are unable to provide POTC have the infrastructure and/or facilities for TEG?
Line 163 – 170: This is a very good justification for TEG. You might think about presenting this data of “normal PT and APTT” in Section 3.3.1.
Figure 1: This is an excellent figure and demonstrates the value and simple interpretation that can be gained from TEG. However, I think you need to provide a better explanation of the metrics used in TEG: R, K, alpha angle, MA and LY30. There should also be a section that tells the reader what deviations indicate as well as the course of treatment based on these deviations.
Section 3.3.3: I am not sure how this section fits into the paper or is it part of Section 3.4.1? Are you showing the reader that antivenom is not an optimal treatment due to the misidentification of species envenomation? Or are there other reasons that you have referenced as to the controversy of using antivenom for treatment. The subject of antivenom should be better developed to explain why this is not as good for prognosis as TEG.
Figure 2: I do not see any pharmacological intervention in your algorithm, e.g. when there is an increase in LY30 (>0-8%) indicating an increase in fibrinolysis, the indication would be an antifibrinolytic like Amicar. Drug therapy plays a critical role in patient care.
Author Response
Thank you for providing helpful comments. Your interest in this article and your constructive criticisms are much appreciated. I worked with all my coauthors in trying to address to every comment of yours, which led to a significant change from the original manuscript.
I answered all your points as below. The changes that are made in the article are highlighted as a yellow color. Please review this revised version and inform us of your decision. I am looking forward to your reply. My answer to your comment is followed by an arrow. Thank you very much for your consideration.
Lines 40 & 41: The sentence beginning with “First, snake venom…” is structurally awkward.
--> We changed this sentence as below.
First, snake venom is composed of various toxins [4,9].
Lines 42 – 44: The word various has been used repeatedly; rewrite to make the concept of venom complexity clearer.
--> We changed this sentence as below.
Snake venoms have different compositions depending on the families and species of the snakes. Different toxins lead to different clinical manifestations such as neurotoxicity, myotoxicity, or coagulotoxicity.
Line 47: Replace showed quiet with “have shown quite.”
--> We made this change.
Line 67 – 68: The sentence beginning with “VICC was commonly…” is structurally awkward.
--> We changed this sentence as below.
VICC was frequently manifested in Asian vipers’ envenomation
Section 3.3.1: Although PT and aPTT are limited in the diagnostics you report, are they adequate to demonstrate a coagulopathy from the snake bite indicating a need for treatment? If so, wouldn’t this lead the attending physician to order further diagnostics?
--> The results of PT and aPTT can guide the treatment in patients with VICC. However, this laboratory methods have reflected relatively late time in terms of developmental stage, which was described this in 3.3.1 Section. This is one of the important reasons why new test or strategy is necessary for the management of VICC.
Line 143: What “some test” are you referring to that requires significantly longer time for completion?
--> We modified the sentence to further clarify its meaning.
First, TEG takes less time than the analysis of coagulation factors. Platelets, PT, and aPTT usually take shorter time and are carried out in most facilities. However, fibrinogen and fibrin degradation products take a lot longer, and they may be unavailable for testing in some facilities.
Line 147 – 148: Do facilities that are unable to provide POTC have the infrastructure and/or facilities for TEG?
--> We mentioned the POCT for TEG. TEG is performed as POCT in ER or OR. We modified the sentences.
Some facilities perform TEG as a point-of-care test (POCT) to make a rapid decision of resuscitation in emergency situations. However, frequent and precise quality control is necessary for TEG POCT .
Line 163 – 170: This is a very good justification for TEG. You might think about presenting this data of “normal PT and APTT” in Section 3.3.1.
--> We added several sentences to this paragraph, as you suggested.
However, failure to detect VICC in the early stage by PT and aPTT has been reported in other studies. The PT and aPTT levels are initially within the normal range and then changed to abnormal levels over a few days, showing clinical manifestations of VICC. Diagnosis of coagulopathy using PT and aPTT alone has many limitations.
Figure 1: This is an excellent figure and demonstrates the value and simple interpretation that can be gained from TEG. However, I think you need to provide a better explanation of the metrics used in TEG: R, K, alpha angle, MA and LY30. There should also be a section that tells the reader what deviations indicate as well as the course of treatment based on these deviations.
--> We added a figure to provide an explanation as you suggested, and we changed legend.
Section 3.3.3: I am not sure how this section fits into the paper or is it part of Section 3.4.1? Are you showing the reader that antivenom is not an optimal treatment due to the misidentification of species envenomation? Or are there other reasons that you have referenced as to the controversy of using antivenom for treatment. The subject of antivenom should be better developed to explain why this is not as good for prognosis as TEG.
--> We understand your concerns. Antivenom is known to be the key treatment of envenomation. Identification of snake species is necessary for accurate diagnosis and administration of an adequate antidote. Monovalent antivenom is the best treatment if the culprit snake is known to the physician. In a lot of cases, it is hard to know what the exact species was, and this is why polyvalent antivenom is usually administered. Immunoassay tests of blood, urine, or urine samples might be helpful for the identification of the culprit snake [38,39]. However, it is difficult to apply these tests in all regions due to limitations such as cost and false-positive results. We added one sentence in this paragraph as below.
Identifying the species of snake can make usage of monovalent antivenom in some regions [38,39]. However, identifying the species of snake by the information from the witness is uncertain [39]. Thus, polyvalent antivenom is commonly administered to patients with a snakebite who need an antivenom.
Figure 2: I do not see any pharmacological intervention in your algorithm, e.g. when there is an increase in LY30 (>0-8%) indicating an increase in fibrinolysis, the indication would be an antifibrinolytic like Amicar. Drug therapy plays a critical role in patient care.
--> We added some sentences of recommended pharmacological interventions in the last paragraph and related references.
Based on the results of the previous studies regarding a TEG-based transfusion algorithm, one can consider transfusing FFP when R time is > 10 minutes. Further, one can consider transfusing platelets when MA is < 50 mm. LY 30 > 3% is critical for the initiation of antifibrinolytic therapy, where one can consider administering an antifibrinolytic agent such as ε-aminocaproic acid or tranexamic acid in patients with VICC.
Round 2
Reviewer 2 Report
The manuscript is improved however there are still English language and general editing issues.
In your response to my initial review you stated that Line 18 had been changed but it has not been changed.
The following lines have sentences that do not make sense or are grammatically incorrect.
Lines 139-140
Lines 140-141
Line 150
Lines 166-167
Llines 172 - 173
Line 176
Lines 261-262
There are two lines that contain the exact same content. I assume a cutting and pasting error. Lines 286-287 same as Lines 277-278.
Lines 301-302 - Again, may be a bit strong. Consider recommending that TEG be added to more traditional testing to further test its usefulness in guiding the treatment of VICC.
Author Response
Thank you for providing helpful comments. Your interest in this article and your constructive criticisms are much appreciated. I answered all your points as below. The changes that are made in the article are highlighted as a yellow color. Please review this revised version and inform us of your decision. I am looking forward to your reply. My answer to your comment is followed by an arrow. Thank you very much for your consideration.
Reviewer 2
In your response to my initial review you stated that Line 18 had been changed but it has not been changed.
- We changed the sentence as you recommended.
Therefore, we believe that TEG may be able to be used to guide hemostatic resuscitation in victims of VICC. (Line 18-19 in the revised manuscript)
The following lines have sentences that do not make sense or are grammatically incorrect.
Lines 139-140
- We changed the sentence as below.
Thromboelastography (TEG) has been widely used during transfusion in patients with trauma as well as during surgeries. (Line 142-143 in the revised manuscript)
Lines 140-141
- We changed the sentence as below.
TEG analyzer consists of a torsion wire, a pendulum, and a disposable cup to hold the sample of whole blood. The cup is rotated back and forth through an angle of 4.45°, six times a minute, to imitate sluggish venous flow, for the whole duration of the test. A pin attached to the torsion wire is inserted into the cup with the whole blood sample. As the coagulation cascades progress, the pin produces electrical signals. Typical waveforms regarding the speed and strength of clot formation are displayed as signals in TEG, and the measurements are transferred onto the computer. (Line 143-149 in the revised manuscript)
Line 150
- We changed the sentence as below.
Second, information that cannot be obtained from PT and aPTT can be gathered. (Line 157-158 in the revised manuscript)
Lines 166-167
- We changed the sentence as below.
Early stages of VICC can be determined using TEG, even if the PT and aPTT remain normal. (Line 173-174 in the revised manuscript)
Llines 172 – 173
- We changed the sentence as below.
TEG may be able to detect VICC presenting procoagulation and fibrinolysis by degradation or fragile fibrin formation. (Line 179-180 in the revised manuscript)
Line 176
- We changed the sentence as below.
C. rhodostoma and H. hypnale have strong procoagulant components and can decrease the clotting time [3]. However, they also have pseudo-procoagulant effects, which can weaken fibrin and thus lead to an anticoagulation state. (Line 183-185 in the revised manuscript)
Lines 261-262
- We changed the sentence as below.
Unlike these studies, other studies have reported that transfusion of coagulation factors may be effective in patients with VICC. (Line 272-274in the revised manuscript)
There are two lines that contain the exact same content. I assume a cutting and pasting error. Lines 286-287 same as Lines 277-278.
- We did mistake as you mentioned. We removed the repeated sentence.
Lines 301-302 - Again, may be a bit strong. Consider recommending that TEG be added to more traditional testing to further test its usefulness in guiding the treatment of VICC.
- We modified the sentence as below.
Further randomized, controlled researches are needed, in particular, to determine the efficacy of TEG in guiding the treatment of VICC. (Line 314-316 in the revised manuscript)
Reviewer 3 Report
This manuscript has had major English revisions and is much better. Very nice job and good luck.
Author Response
Thank you!